# Aflatoxin B1 Detoxification Potentials of Garlic, Ginger, Cardamom, Black Cumin, and Sautéing in Ground Spice Mix Red Pepper Products

**DOI:** 10.3390/toxins15050307

**Published:** 2023-04-24

**Authors:** Tadewos Hadero Medalcho, Kebede Abegaz, Engida Dessalegn, Juan Mate

**Affiliations:** 1School of Nutrition, Food Science and Technology, College of Agriculture, Hawassa University, Hawassa P.O. Box 05, Ethiopia; 2Hawassa College of Teacher Education, Hawassa, Ethiopia; 3Public University of Navarra (UPNA), 31006 Navarra, Spain

**Keywords:** antioxidants, AFB1 detoxification, major spices, phytochemicals, mixed red pepper, sauté

## Abstract

The uses of natural plant origin bioactive compounds are emerging as a promising strategy to detoxify aflatoxin B1 (AFB1). This study aimed to explore the potential of cooking, phytochemicals content, and antioxidant activities derived from garlic, ginger, cardamom, and black cumin to detoxify AFB1 on spice mix red pepper powder (*berbere*) and sauté. The effectiveness of the samples was analyzed for AFB1 detoxification potential through standard methods for the examination of food and food additives. These major spices showed an AFB1 level below the detection limit. After cooking in hot water for 7 min at 85 ℃, the experimental and commercial spice mix red pepper showed the maximum AFB1 detoxification (62.13% and 65.95%, respectively). Thus, mixing major spices to produce a spice mix red pepper powder had a positive effect on AFB1 detoxification in raw and cooked spice mix red pepper samples. Total phenolic content, total flavonoid content, 2,2-diphenyl-1-picrylhydrazyl, ferric ion reducing antioxidant power, and ferrous ion chelating activity revealed good positive correlation with AFB1 detoxification at *p* < 0.05. The findings of this study could contribute to mitigation plans of AFB1 in spice-processing enterprises. Further study is required on the mechanism of AFB1 detoxification and safety of the detoxified products.

## 1. Introduction

Foodborne pathogens are the cause of food safety and quality impairment and major public health problems across the world. Spices are among the most exposed, easily contaminated in a wide range and are susceptible to contaminating fungi at different geographical locations [1]. Inappropriate postharvest handling practices such as uncontrolled/delayed drying, or moisture level allowed to exceed critical values facilitates mold growth and mycotoxins production during the processing or storage of the products [2]; particularly, in countries with tropical climates that have high temperature, humidity, and rainfall where spices are usually produced [2,3]. Aflatoxins (AFs) are considered to be the most important mycotoxins and have a wide occurrence in spices, cereals, oils, fruits, vegetables, milk, and meat [3]. Aflatoxin B1 (AFB1), one of the well-known AFs (AFB1, AFB2, AFG1, and AFG2), is a frequently occurring and toxicologically recognized group of Afs in spices including red pepper (*Capsicum annum* L.) [3,4]. Aflatoxin B1 (AFB1) is the most harmful and is responsible for more than 75% of all Afs in food including spices and feed contamination, which is produced by *Aspergillus flavus* and *Aspergillus parasiticus* [5]. It is capable of producing acute toxicities in mammalians at the level of chronic exposure including immunosuppression, genotoxic, carcinogenic, mutagenic, neurotoxic, teratogenic, and hepatotoxic effects [6]. The International Agency for Research on Cancer (IARC) has categorized AFB1 as Group I carcinogenic to humans, while other AFs are less toxic (in descending order; B1 > G1 > B2 > G2) [7]. 

Red pepper is extensively used for flavor and as a colorant but is sensitive to AF contamination depending on atmospheric temperature, humidity, insect damages, and drying and processing conditions. Sun drying is a common postharvest handling practice in some countries, which involves spreading the bulk of pepper pods on the ground in a mass that supports contamination with fungi [8]. In Ethiopia, pepper pod drying and spice mix red pepper powder (*berbere*) production methods are not well controlled, which may create a favorable condition for aflatoxigenic mold growth. Moreover, poor postharvest handling and a sprinkling of water during trade practices and processing to keep the color and avoid the burning sensation of red pepper pods during cutting peduncles may also contribute [9]. Processed pepper products, for instance crushed pepper, ground pepper, and paprika/capsicum, are the main ingredients of Ethiopian cuisine such as stew, hot sauté, and paste [10], but they are more susceptible to AFs than fresh pods [9]. 

Aflatoxin detoxification refers to those postharvest treatments directed to eliminate, diminish, or inactivate to a safe level by applying pre-intake detoxification technologies such as physical, chemical, or biological treatment [11,12]. Aflatoxin B1 detoxification using natural plant products is currently receiving research attention because it is simple and easy to be obtained, biodegradable, eco-friendly, renewable, and technically and economically feasible [4,13]. The AFs, particularly AFB1 decontamination by herbs and/or spices, may rely partly on polyphenolic compounds such as phenolic acids, flavonoids, carotenoids, and organosulfides of plants origin on their concentration [12,14]. Physical, chemical, and biological treatment methods have been investigated to prevent the growth of AFs producing fungi and degrade or detoxify AFs levels in foods and feeds. Great success has been achieved to reduce mycotoxigenic fungi and mycotoxins in foods using plant extracts and plant essential oils [12,15]. Several medicinal herbs, spices, and higher plants have shown inhibitory effects on the growth of toxigenic fungi and the production of toxins [16,17]. Chemical transformation of the AFB1 parent compound to another nontoxic compound, most likely lacking a lactone ring, is proposed as a possible mechanism of AFB1 degradation (Figure 1) [4,11].

However, thermal processing demonstrated diverse effects on the bioactive phytochemicals and functional properties of herbs and spices [18]. In many whole spices including pepper, processing changes their chemical compounds and proportions to varying degrees, often giving rise to different flavor profiles [19] and decreasing proteins solubility and digestibility [20]. Employing severe heat during cooking may result in important physical and chemical changes that negatively affect the sensory quality and nutritional value [21], and loss of volatile oils, nutrients, and phytochemicals in spices [22]. On the other hand, the effect of cooking on antioxidant activity is not always consistent, which may be associated with the nature of the food matrix, the type of cooking method used, and the cooking time-temperature combination [23]. Similarly, stewing increased antioxidant activity, probably because of heat-liberating antioxidant compounds [24]. 

In general, the flavoring components as well as extractable bioactive compounds of herbs and spices are mostly reduced after cooking, which might be due to the leaching out of phytochemicals into cooking water [25]. However, cooking with water enhances the extraction of bound phytochemicals and thereby increases the phytochemicals and antioxidant activities of herbs and spices. This is true if and only if leached-out bioactive compounds are considered (not decanted/lost) during cooking. There is no previous report on the application of ground spices for the detoxification of AFB1 in contaminated foods and food additives, whereas some studies revealed the efficacy of using plants incorporated in foods such as additives as an alternative method for preventing natural oxidants and antimicrobial activities [26]. Thus, mixing ground spices during spice mix red pepper powder (*berbere*) production and cooking the mix is predicted to detoxify AFB1. Therefore, this study investigated the AFB1 detoxification potential of the major spices (garlic, ginger, cardamom, and black cumin) in a spice mix red pepper powder and sauté. 

## 2. Results

### 2.1. Aflatoxin B1 Level and Detoxification 

In the present study, the AFB1 level of five major ground spices (garlic, ginger, cardamom, and black cumin), two ground spice mix red pepper, and two spice mix red pepper sauté was analyzed through the extraction process. In addition, AFB1 detoxification potentials of garlic, ginger, cardamom, and black cumin in spice mix red pepper powder and sauté were analyzed. The linearity was checked for AFB1 standard solution of six concentration levels in the concentration range from 0.0 to 1.2 µg kg^−1^. The calibration curve was linear with correlation coefficients of R^2^
*=* 0.987 and the limit of detection (LOD) was below ≤2 µg kg^−1^. Thus, the method was suitable for the simultaneous determination of AFB1.

The concentration of AFB1 of the duplicate samples of RP, experimental and commercial spice mix red pepper and sauté were shown in Figure 2. The results revealed that AFB1 level in RP (24.79 µg kg^−1^) was significantly (*p* < 0.05) higher than all red pepper samples. Conversely, the AFB1 level of commercial spice mix red pepper sauté (CS) (8.44 µg kg^−1^) was significantly (*p* < 0.05) lower than the entire red pepper samples. 

The initial AFB1 level in uncooked ground RP was 24.79 µg kg^−1^ when mixing it with ground dietary spices and sautéing significantly detoxified AFB1. A statistical analysis showed a very significant (*p* < 0.05) detoxification range in AFB1 from uncooked EP (38.98%) to cooked ES (62.13%) and from uncooked CP (23.47%) to cooked CS (65.95%). In general, percent AFB1 detoxification was significantly (*p* < 0.05) higher in cooked than uncooked spice mix red pepper samples (Table 1). 

### 2.2. Total Phenolic and Total Flavonoid Contents, and Antioxidant Activities of Entire Samples

The total phenolic content (TPC) and total flavonoid content (TFC) of entire samples are presented in Figure 3. Those phytochemicals not correlated (data not shown) with percent AFB1 detoxification at *p* < 0.05 were excluded. The TPC of major spices ranging from 12.77 to 13.13 mg GAE g^−1^ showed no difference between garlic and ginger, and cardamom and black cumin. The TFC of black cumin, ginger, garlic, and cardamom, respectively, 12.92, 12.73, 12.65, and 12.43 mg QE g^−1^, were significantly (*p* < 0.05) different. On the other hand, TPC of RP, EP, ES, CP, and CS was recorded as 10.43, 13.62, 14.21, 13.76, and 13.96 mg GAE g^−1^, respectively, but ES, EP, and RP showed a significant difference at *p* < 0.05. Similarly, TFC ranged from 13.21 to 15.14 mg QE g^−1^ in ascending order: RP, EP, ES, CP, and CS with significant (*p* < 0.05) differences with each other. 

Figure 4 shows the 2,2-diphenyl-1-picrylhydrazyl (DPPH), ferric ion reducing antioxidant power (FRAP), and ferrous ion chelating activity (FICA) of entire samples correlated with percent AFB1 detoxification at *p* < 0.05. The antioxidant assays not correlated (data not shown) with percent AFB1 detoxification at *p* < 0.05 were excluded. The IC50 (inhibition concentration) of garlic, ginger, cardamom, and black cumin able to scavenge 50% of the DPPH activity, respectively, was 133.90, 159.43, 137.96, and 215.81 µg mL^−1^ with significant (*p* < 0.05) differences except garlic and cardamom. The FRAP antioxidant activity of ginger, garlic, black cumin, and cardamom was recorded in ascending order: 98.52, 96.55, 89.98, and 87.93 mg TE g^−1^ with significant difference at *p* < 0.05. On the other hand, FICA ranged from 20.25 to 21.92 mg QE g^−1^, respectively in descending order: garlic, black cumin, ginger, and cardamom, but garlic was significantly (*p* < 0.05) different from the others. The IC50 value of DPPH of control and spice mix uncooked red pepper, and sauté ranged from 31.29 to 189.08 µg mL^−1^ in ascending order: CS, CP, ES, EP, and RP with significant differences at *p* < 0.05. The FRAP antioxidant capacity was 91.12, 92.34, 93.71, 93.77, and 94.68 mg TE g^−1^, respectively for RP, EP, CP, ES, and CS; where CS showed significant (*p* < 0.05) difference with RP and EP. The value of FICA was 23.33, 24.08, 24.77, 25.23, and 25.40 mg QE g^−1^ for RP, EP, CP, CS, and ES, respectively; except RP, the others showed no difference.

### 2.3. Correlation Analysis

The correlated TPC, TFC, and antioxidant assays (DPPH, FRAP, and FICA) versus percent AFB1 detoxification at *p* < 0.05 are shown in Table 2. Statistically, a comparison of the quantitative analysis of TPC (r = 0.9), TFC (r = 0.68), DPPH (r = 0.87), FRAP (r = 0.82), and FICA (r = 0.86) showed a significant (*p*
< 0.05) positive correlation with percent AFB1 detoxification in raw and cooked spice mix red pepper products. A positive direct correlation between TPC, TFC, DPPH, FRAP, and FICA against percent AFB1 detoxification led to a decrease in the AFB1 level in raw spice mix red pepper powder. The synergistic effect of TPC, TFC, DPPH, FRAP, and FICA with water (used for cooking) played a major role in ABF1 detoxification in sautéing. 

## 3. Discussion

Several recent studies revealed that garlic, ginger, and black cumin have specific bioactive compounds with antibacterial, antiviral, antifungal, antioxidant, and preservative properties [27,28,29]. Despite the widespread uses of Ethiopian cardamom, there is scarce information regarding antimicrobial activities. However, some reports were presented from related genera on antimicrobial, antifungal, antibacterial, and antiviral activities in *Aframomum* species such as *A. giganteum*, *A. melegueta*, and *A. citratum* [30]. This might be attributed to the presence of phenolics and flavonoids in major spices, which agrees with the previous work of Gupta [31]. The method showed good linearity with the correlation equation, y = 1.5125x + 1.1351 as exhibited by the square of the correlation coefficient, R^2^ = 0.9901 (Figure 5). The LOD was used to determine AFB1 because it was below ≤2 µg kg^−1^. The AFB1 level of all duplicate samples of the major spices used for EP powder production was below the LOD (≤2 µg kg^−1^) (data not shown) in contrast to a tropical climate, which is ideal for fungal growth, and mycotoxin production may be attributed to the samples collection period (harvesting time) and drying conditions. A similar report of previous study was presented in black pepper in Korea [32]. Therefore, they may not have any risk to public health. 

The AFB1 level of ground red pepper reported in Turkey (77.13 µg kg^−1^) [5], in Nigeria (156 µg kg^−1^) [8], and in Ethiopia (average 312.5, and 50.5 µg kg^−1^, respectively in 1996 and 2001 by Habtamu and Kelbessa) as cited by Mekuria [33] and Tariku et al. [34] was higher than the present finding in ground red pepper (24.79 µg kg^−1^). Similarly, Pickova et al. [35] reported an AFB1 level of 35 µg kg^−1^ based on the Rapid Alert System for Food and Feed (RASFF) database of 2015–2019 that was higher than the current finding. However, the AFB1 level in ground red pepper reported by Barani [36] in Iran (15.51 µg kg^−1^) was lower than the current finding. The previous work reported on AFB1 level by Thanushree et al. [37] in spice mixes purchased from different cities in Malaysia, and Syamilah et al. [38] in curry powder (mix of coriander seeds, cumin, turmeric, and ginger), respectively 14.36 µg kg^−1^ and 2.26 µg kg^−1^, were lower than the present finding on spice mix red pepper samples. These contradictory results may be due to the variety of the spice crops, different sample collection and preparation (sorting, cleaning, and milling) techniques, sampling location, and handling conditions (drying and storage) [39,40,41]. Most of the countries that reported high levels of AFB1 contamination in spices have a tropical climate, where the temperature, humidity, and rainfall provide optimal conditions for the growth of the mold [32].

In the current study, control and spice mix red pepper products showed an AFB1 level about 20% and 100%, respectively, above the maximum permissible limit of the USA (20 μg kg^−1^) and European Commission (5 μg kg^−1^) for spices [42]. However, Ethiopia has not established a residue limit of AFB1 in spices. This may be unfit for both consumers and for trade in European and US American markets. It was assumed that the reason for the high levels of AFB1 in the present study might be due to the difference in red pepper pods marketing sources, sorting, grading, or cleaning and gradual drying, and poor storage facilities of spices processing enterprises.

The AFs toxicity data demonstrated that the presence of a double bond in the terminal furan ring that may undergo reduction reaction and to a lesser extent the lactone rings are key factors for its toxic and carcinogenic activities [14,43,44]. Therefore, the detoxification of AFB1 may be due to the removal of the double bond in the terminal furan ring and the modification of the lactone ring leading to the conversion of the parent compound into several products (such as phenolic and acidic groups) of much lower toxicity [11,14]. The existence of water-soluble extractable phytochemicals such as phenolics in the mixed spice red pepper sauté may be responsible for the alteration and breakage of the molecular structure of phytochemicals [17,45]. Aflatoxins detoxification potentials of plant products were reported by several authors at different times [4,17,26,46]. Previous studies have shown that certain herbs and spices used in food production and cooking showed detoxification of AFB1 in fruits and vegetables, including red pepper and other foodstuffs [47]. Jalili [48] reported that the decontamination of food by heating in the presence of water is easier and more effective. 

The present study finding was similar to the finding of Ponzilacqua et al. [26], who reported that AFB1 detoxification by using *Rosmarinus ofcinalis*, *Origanum vulgare*, and *Psidium cattleianum* exhibited 60.3%, 38.3%, and 30.0%, respectively, and Misgana and Alemayehu [4] using aqueous extracts of *Allium sativum* and *Citrus lemon* revealed 61.7% and 56%, respectively. However, Iram et al. [45] reported 80% and 95.21% AFB1 detoxification using Ajwain (*Trachyspermum ammi*) seeds extract and leaves extract of *Corymbia citriodora*, respectively, which was higher than the present finding. Similarly, Vijayanandraj et al. [17] and Rushing and Selim [49] reported 98.4%, and 90.4% AFB1 detoxification, respectively, using leaves of Vasaka (*Adhatoda vasica Nees*), and basil (*Ocimum basilicum*) extracts. Conversely, Karlovsky et al. [47] reported 34% AFB1 detoxification during normal cooking with water that was lower than the present finding. The AFB1 detoxification in the present study might be due to the synergistic effect of cooking time-temperature combination (7 min and 85 °C) and water content, and type and amount of spices used for the production of spice mix red pepper powder.

In the present study, TPC and TFC were increased for both uncooked and cooked spice mix red pepper samples, which agreed with the findings of Inchuen et al. [50] and Shaimaa et al. [51] who reported that TPC and TFC of Thai red curry paste and sweet and chili pepper, respectively, increased after boiling. This might be due to disruption of plant cell walls during heat treatment that resulted in better extractability of phenolic compounds [18,23,50]. This is true if and only if decanting is not involved as a method of cooking [52]. According to the report of Bruck et al. [53] and Chan et al. [54], the antioxidant activities were highly associated with the presence of total flavonoid and total phenolic compounds. Generally, the current study result showed an increase in antioxidant activities that may be due to a loosening of antioxidant moieties, and improved extractability of the antioxidant compounds such as phenols from the samples [18,23].

Most of the previous studies conducted on the use of plant extracts for aflatoxin control focused on the ability to inhibit the growth of the primary aflatoxigenic fungi (*A. flavus* and *A. parasiticus*) and the decontamination of toxins. However, the present study mainly focused on major spices and sautéing examination for their ability to detoxify AFB1 in spice mix red pepper products. Misgana and Alemayehu [4] and Altemimi et al. [55] reported similar correlation of some plant extracts (such as thyme, garlic, ginger, and lemon) against AFB1 decontamination. According to Guldiken et al. [56] report, there is a positive correlation between the antimicrobial activities and phenolic content of spices and medicinal herbs. The results of correlation analysis between total phenols and antioxidants with antimicrobial activity indicated a positive relation with the plant extracts [57,58]. Therefore, in the present study, the correlation analysis data suggested that mixing ground major spices that are rich in TPC, TFC, and antioxidant activities with ground red pepper for the production of spice mix red pepper powder and cooking the mixed product with water play a key role in detoxification of AFB1. 

Aflatoxins are stable up to their melting points (≥250 °C) when heated without water [47,59]; thus, they are quite resistant to ordinary thermal food processing [48]. The finding of this study suggested that mixing spices during spice mix red pepper powder production, and water assisted cooking could be used to effectively detoxify AFB1 in foods and food additives. Thus, major spices used for the production of EP powder, and sautéing were sufficient to observe a significant difference in ABF1 level between the treated and non-treated samples. As a result, the current finding is consistent with the existing literature and adds value by giving credit to the indigenous knowledge on the use of spices for AFB1 degradation in line with their application to food organoleptic quality improvement and preservation. 

## 4. Conclusions 

Spice mix red pepper powder is a well-known spice powder and its sauté is the key ingredient of Ethiopian stew. To the best of our knowledge, this is the first report of detoxification of AFB1 by mixing ground red pepper with ground garlic, ginger, cardamom, and black cumin. The percent AFB1 detoxification of uncooked spice mix red pepper was lower than the cooked one. Thus, there is a synergistic effect between mixed spices and cooking mixed spices with water on AFB1 detoxification. Ground garlic, ginger, cardamom, and black cumin alone or as a mixture, possess a high phytochemicals content and antioxidant activities in spice mix red pepper powder. A good positive correlation was revealed between TPC, TFC, DPPH, FRAP, and FICA against percent AFB1 detoxification in spice mix red pepper powder and sauté. The current study findings could contribute to the development of reclamation strategies and safe technologies for AFB1 detoxification in various spices processing industries. Therefore, further studies need to be undertaken for the remaining spices to optimize mixing proportions, and types of spices used for the production of spice mix red pepper powder. Moreover, more study is recommended on the cooking time-temperature combination of Ethiopian sauté, and for structural clarification and mechanism of interaction between AFB1 and bioactive compounds of major spices responsible for observed detoxification together with the effects of detoxified AFB1 products on humans health.

## 5. Materials and Methods

### 5.1. Experimental Design 

The study employed a preliminary survey as well as a laboratory experiment. The preliminary survey part of the study included visiting different open markets and spices processing enterprises and interviewing to explore potential areas and varieties. The enterprises were purposively selected for sample collection, which helped to avoid the tractability problem of the sample source and varieties of the spices, particularly red pepper pods. Moreover, it helped to minimize traders’ fraudulent behavior of mixing high-priced red pepper pods with low-priced ones from different sources and varieties. Simple random sampling was applied to collect samples from purposively selected enterprises to minimize unbiased representation of the total population. The laboratory analysis involved analyzing different parameters of the collected spice samples and their mixture products. The experiment was conducted using completely randomized design (CRD) with two replicates. The control red pepper (RP) sample was a recognized control that helped to evaluate whether the experiment was performed properly or not, while the spice mix red pepper (*berbere*) sample was collected from processing enterprises called commercial spice mix red pepper (CP) which was used to make the distinction between control red pepper and experimental spice mix red pepper (EP) samples. Ground garlic (*Allium sativum*), ginger (*Zingiber officinale*), cardamom (*Aframomum corrorima* (Braun) *p*. C. M. Jansen), and black cumin (*Nigella sativa* L.) samples were blended with ground red pepper for the production of spice mix red pepper powder based on the practices of Ethiopian medium and small-scale spices processing enterprises with slight modifications. 

### 5.2. Sample Collection and Experimental Sauté Preparation 

Fresh garlic cloves, ginger rhizomes, and cardamom pods, and dry red pepper pods and black cumin seeds, respectively 12, 12, 9, 6, and 3 kg, were purchased from purposively selected Hawassa city medium and small-scale spices processing enterprises. Entire samples were collected between February to March 2022. Each sample of red pepper, garlic, ginger, cardamom, and black cumin was mixed manually to make representative samples. All samples were properly cleaned and sun dried at Hawassa University College of Agriculture, School of Nutrition, Food Science and Technology Laboratory. 

The dried and cleaned garlic, ginger, cardamom, black cumin, and red pepper samples were individually ground into a fine powder with a high-speed multifunction comminutor (400A, RRH, China), and sieved with 500 μm mesh size. Each finely ground sample was divided into two portions; one portion of it was used to mix with each other to produce EP, while the other portion remained as ground sample. A proportionate mixing was done using a high-speed multifunction comminutor (Table 3). The EP powder was divided into two portions; one portion of it was used for experimental sauté preparation, while the other portion remained as EP powder. The finely ground samples were packed in a food grade polyethylene bag and stored in a dry and dark place at room temperature until extraction. 

The Ethiopian stew (such as *shiro*, lentil stew, potato stew) preparation process begins with sautéing, in which preliminary ingredients such as fresh chopped onion and garlic, and oil were used. However, in this experimental sauté preparation, those preliminary ingredients were excluded to avoid their effect on the phytochemicals content and antioxidant activity of garlic, ginger, cardamom and black cumin. Therefore, the experimental sauté was prepared by mixing 50 g of EP powder and 400 mL boiling water at 65 °C. It was simmered at 85 ℃ within 7 min. The same fashion was applied for CS preparation. The prepared sautés were oven-dried at 60 °C for 48 h and ground into a fine powder with a high-speed multifunction comminutor, and sieved with 500 μm mesh size. It was packed and stored in the same style as the powder samples until extraction. 

### 5.3. Experimental Section

#### 5.3.1. Aflatoxin B1 Determination

Aflatoxin B1 was determined spectrophotometrically using AFB1 kit (Competitive enzyme immunoassay for the quantitative detection of Aflatoxin B1. BioTeZ Berlin, Germany) according to the manufacturer’s instructions [60]. All samples and reagents in the assay kit were kept at room temperature (25 °C) prior to analysis. The fine ground and homogeneous 5 g sample was weighed in a suitable container and mixed with 25 mL of methanol: water (70:30, *v*/*v*). This suspension was shaken intensively for 3 min to extract the aflatoxin. The solids were briefly settled before filtering. The suspension was then filtered via a folded filter pepper for quantitative analysis. The sample extract was diluted in a new container with a 1:10 ratio with the sample diluent (1 part filtrate + 9 parts AFB-SAMPLE-BUF). The dilution factor of 1 mL diluted filtrate or 0.1 g solid sample is equal to 50. The analytical grade reagents were used in this experiment.

A 25 mL measure of a 10-fold concentrated washing solution (WASH-10x) was diluted with 225 mL distilled water. A 50 µL measure of the standards (AFB1-0 to AFB1-5) was pipetted into the corresponding cavities. Similarly, 50 µL of the sample extracts was pipetted into the corresponding cavities. Finally, 50 µL AFB1 peroxidase conjugate (AFB1-HRP conjugate) was added into all cavities. The microplate was covered with adhesive foil and briefly shaken on the microplate shaker. It was incubated for 30 min at room temperature and protected from light. Before washing, all cavities were aspirated or shaken out. Then, 300 µL of reconstituted washing solution was added into each cavity and aspirated out. This process was repeated twice. A 100 µL measure peroxidase substrate solution (3, 3′, 5, 5′-tetramethylbenzidine, TMB) was added into all cavities for the color reaction (forming a blue color). The microplate was covered with adhesive foil and then shaken briefly. It was incubated for 15 min at room temperature and protected from light. Finally, 25 µL of stop solution (STOP-H_2_SO_4_) was pipetted into each cavity, and then the color change was revealed (from blue to yellow). The absorbance of the 96 cavities was measured at 450 nm with a reference wavelength of 620 nm using Thermo Scientific™ Multiskan™ GO Microplate Spectrophotometer (Thermo Fisher Scientific Oy Ratastie, Vantaa, Finland). The sensitivity of the spectrophotometer was determined in terms of linearity and LOD. The method linearity was determined in duplicate with six AFB1 standard concentrations (0.0, 0.05, 0.1, 0.25, 0.5, and 1.2 μg mL^−1^), which was used to draw a standard curve to quantify AFB1 using the regression equation. 

The AFB1 content was expressed as μg g^−1^ using a calibration equation based on the calibration curve (Figure 5). The result was multiplied by the corresponding dilution factor (50) to obtain the actual concentration of the sample. The efficiency of major spices in EP and sauté was calculated in terms of percent detoxification of AFB1 using the following equation.
$$\%\;\mathrm{Detoxification}=\frac{C0-\mathrm{Ct}}{C0}\times 100$$
where;

C0 = the concentration of AFB1 at the beginning (ng mL^−1^) 

Ct = the concentration of AFB1 in the treatment (ng mL^−1^)

#### 5.3.2. Determination of TPC, TFC, and Antioxidant Activities 

The TPC, TFC, and antioxidant activities of major spices, control red pepper, experimental and commercial spice mix red pepper samples were analyzed following standard methods for the examination of food and food additives. Then, the correlation between percent AFB1 detoxification and TPC, TFC, and antioxidant assays were examined in uncooked and cooked spice mix red pepper products. The amount of TPC and TFC were investigated by the Folin–Ciocâlteu method [56,61], and AlCl_3_ colorimetric method [61,62], respectively, with some modifications. Similarly, the antioxidant assays such as DPPH free radicals scavenging activity, FRAP, and FICA were determined using the methods described by Ali et al. [61], Xiaonan et al. [63], and Ali et al. [61], respectively, with some modifications. 

### 5.4. Statistical Analysis 

All data were subjected to one-way analysis of variance (ANOVA), using samples, and sautéing time and temperature combination as independent variables, and the responses were determined as dependent variables. Tukey’s honestly significant differences (HSD) multiple rank tests at *p* < 0.05 was used for the comparison of means value using SAS JMP_14 software (Cary, NC, USA). Correlation analysis was used to evaluate the relations between TPC, TFC, and antioxidant activities against percent AFB1 detoxification. The doublet data results were expressed as mean ± standard error (SE). 

## Figures and Tables

**Figure 1 toxins-15-00307-f001:**
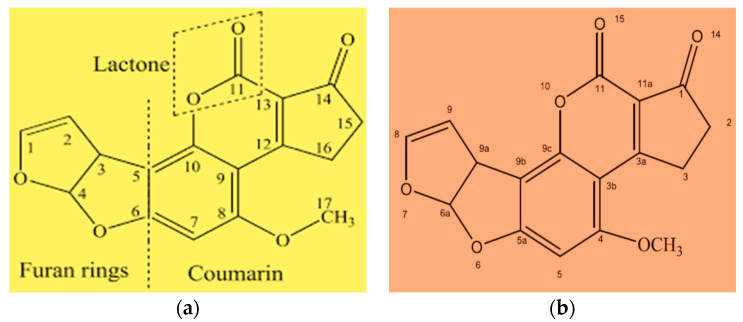
Chemical structure of aflatoxin B1: (**a**) The numbers indicate carbon atoms in AFB1, C_17_H_12_O_6_; and (**b**) the number (1–15) indicates reactive sites of AFB1 [4,11].

**Figure 2 toxins-15-00307-f002:**
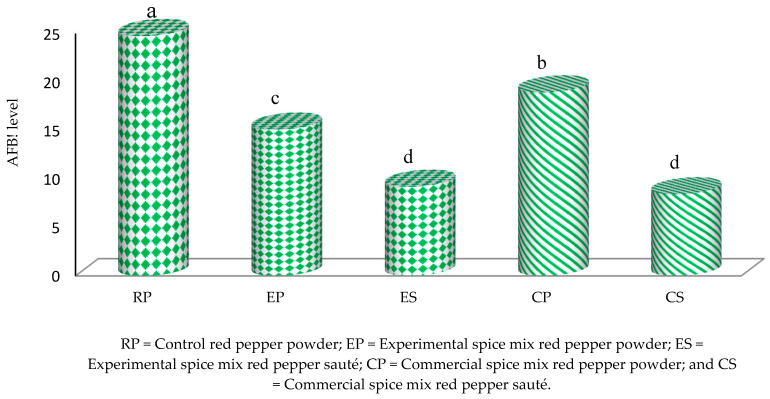
Aflatoxin B1 level of control, experimental and commercial red peppers, and sauté. Levels of mean values not connected in the column by the same letter were significantly different at *p* < 0.05.

**Figure 3 toxins-15-00307-f003:**
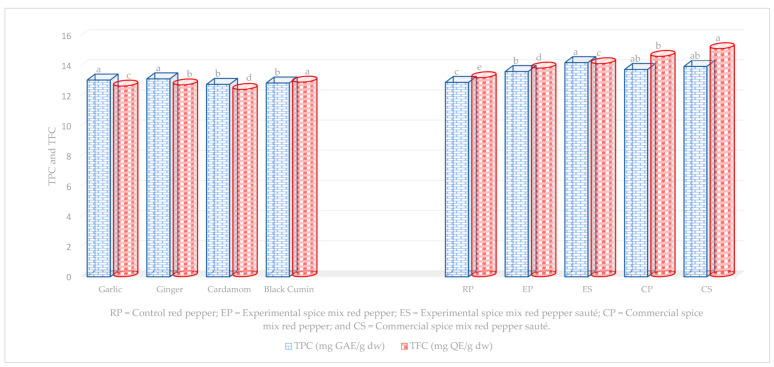
Total phenolic and total flavonoid contents of all samples. Levels of mean values not connected in the column by the same letter were significantly different at *p* < 0.05.

**Figure 4 toxins-15-00307-f004:**
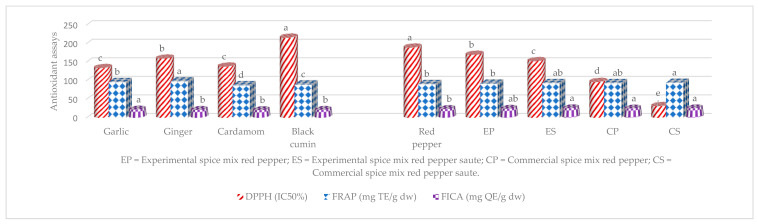
The DPPH, FRAP, and FICA of all samples. Levels of mean values not connected in the column by the same letter were significantly different at *p* < 0.05.

**Figure 5 toxins-15-00307-f005:**
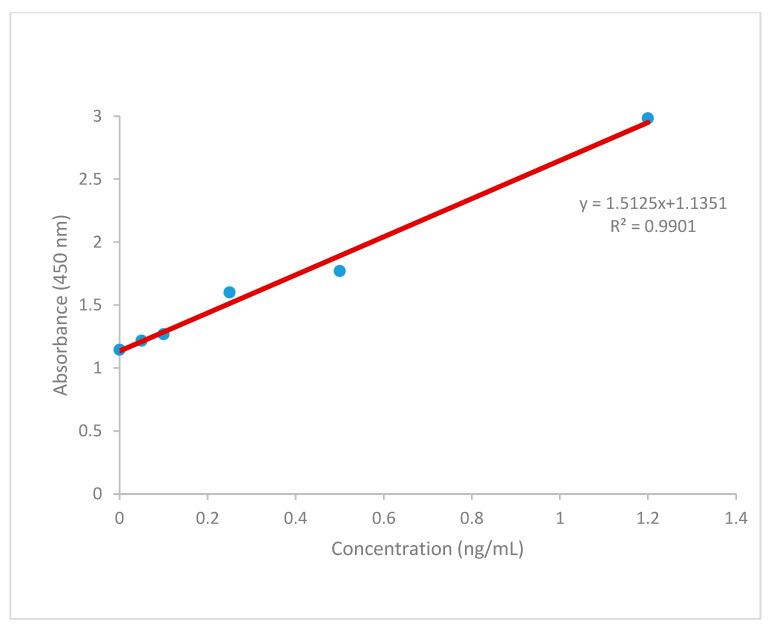
Calibration curve of AFB1 standards.

**Table 1 toxins-15-00307-t001:** Aflatoxin B1 percent detoxification of control, experimental and commercial red pepper, and sauté.

Treatment	% Detoxification ± SE
Control red pepper (RP)	0.00 ± 0.05 ^d^
Experimental spice mix red pepper (EP)	38.98 ± 0.05 ^b^
Experimental spice mix red pepper sauté (ES)	62.13 ± 0.05 ^a^
Commercial spice mix red pepper (CP)	23.47 ± 0.05 ^c^
Commercial spice mix red pepper sauté (CS)	65.95 ± 0.05 ^a^

Levels of mean values not connected in the column by the same letter were significantly different at *p* < 0.05. Least square means differences are separated by Tukey’s HSD test at α = 0.05. SE = Standard error.

**Table 2 toxins-15-00307-t002:** Correlations between TPC, TFC, and antioxidant assays against percent AFB1 detoxification at *p* < 0.05.

Variable	By Variable	Correlation (r)	Count (n)	Signif Prob
Detoxification of AFB1 (%)	TPC (mg GAE g^−1^ dw)	0.9	10	0.0004
Detoxification of AFB1 (%)	TFC (mg QE g^−1^ dw)	0.68	10	0.0305
Detoxification of AFB1 (%)	DPPH free radical scavenging activity (IC50)	0.87	10	0.0008
Detoxification of AFB1 (%)	FRAP (mg TE g^−1^ dw)	0.82	10	0.0037
Detoxification of AFB1 (%)	Chelating potential of FICA (mg QE g^−1^ dw)	0.86	10	0.0013

**Table 3 toxins-15-00307-t003:** Ground ingredients mixing ratio for experimental spice mix red pepper powder and sauté production.

Treatments	RP (%)	GA (%)	GI (%)	CA (%)	BC (%)
Control red pepper (RP)	100	0	0	0	0
Garlic (GA)	0	100	0	0	0
Ginger (GI)	0	0	100	0	0
Cardamom (CA)	0	0	0	100	0
Black cumin (BC)	0	0	0	0	100
Experimental spice mix red pepper (EP)	67.5	13.5	9.5	6.8	2.7
Experimental spice mix red pepper sauté (ES)	67.5	13.5	9.5	6.8	2.7
Commercial spice mix red pepper (CP)	unknown	unknown	unknown	unknown	unknown
Commercial spice mix red pepper sauté (CS)	unknown	unknown	unknown	unknown	unknown

## Data Availability

All relevant data are within the paper.

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
