# Peer review of "Aflatoxin B1 Detoxification Potentials of Garlic, Ginger, Cardamom, Black Cumin, and Sautéing in Ground Spice Mix Red Pepper Products"

_toxins, 2023, doi:10.3390/toxins15050307_

Round 1
Reviewer 1 Report
One remark: It’s a nice study. Are there any food producers involved in this study, so that this information can be used for preparing commercial foods for Humans?
Line 10: food and food ingredient: Does it mean food and feed ingredients?
Line 354: How were the ratios of blends decided?
Line: 406: Why was LC-MS/MS analysis for AFB1 not performed?
Please check the English one more time
Author Response
Dear Sir/ Madam;
I hope you are doing well.
As your comments and concerns, all valuable comments are incorporated, you can see from the attached document. Here is the highlight for your convenience.
Response to Reviewer 1 Comments
- Quality of English Language: English spelling and other punctuations, grammar and sentence construction, the style and paragraph arrangement revisited throughout the whole document.
- Adequate description of methods: The methodology of this experiment is revised and well described as of the aflatoxin B1 kit manufacturer’s instruction, particularly section 5.1 (to some extent), 5.2 (partly), 5.3 as a whole and the sequential order of this sub tile is reorganized as 5.3.1 changed to 5.3.2, and vice versa.
- Concerning involvement of food producers in this study: Ethiopian small and medium scale spices processing enterprises were involved in this study during preliminary survey, so they may apply this information for preparing commercial food additives for humans.
Response 1 (Point 1): Line 10: Concerning food and food ingredients: re-written as ‘food and food additives’.
Response 2 (Point 2): Line 354: Concerning the ratio of blends: The blend formulation was made based on Ethiopian small and medium scale spices processing enterprises and the tradition of the community with slight modification.
Response 3 (Point 3): Line: 406: This experiment was conducted using Thermo Scientific™ Multiskan™ GO Microplate Spectrophotometer. It is used to perform photometric research applications such as DNA and RNA quantification and purity, protein assays, enzyme assays, kinetic assays, immunoassays, cell proliferation and cytotoxicity assays, apoptosis assays, reporter gene assays, GPCR assays. Moreover, free choice of wavelengths from 200 to 1000 nm for the demands of various assays. Besides, the device availability is also a matter.

Reviewer 2 Report
This manuscript investigated phytochemicals content and antioxidant activities derived from garlic, ginger, cardamom, and black cumin, and detoxication of AFB1 on spice mix red pepper powder and sauté. The study designed well and provided some interesting information. However, some comments for the manuscript need to be addressed before it can be accepted.
In the section of Introduction (lines 34-42), detrimental properties of AFs should be introduced such as carcinogenic, teratogenic, mutagenic, nephrotoxic and hepatotoxic properties, and allow me to suggest the following publications to be cited herein, please.
Wang, Y.; Liu, F.; Zhou, X.; Liu, M.; Zang, H.; Liu, X.; Shan, A.; Feng, X. Alleviation of Oral Exposure to Aflatoxin B1-Induced Renal Dysfunction, Oxidative Stress, and Cell Apoptosis in Mice Kidney by Curcumin. Antioxidants 2022, 11, 1082. https://doi.org/10.3390/antiox11061082
Loncar, J.; Bellich, B.; Cescutti, P.; Motola, A.; Beccaccioli, M.; Zjalic, S.; Reverberi, M. The Effect of Mushroom Culture Filtrates on the Inhibition of Mycotoxins Produced by Aspergillus flavus and Aspergillus carbonarius. Toxins 2023, 15, 177. https://doi.org/10.3390/toxins15030177
Line 59: “easy to obtain” should be “easy to be obtained”
Line 63: add the punctuation “.” before “Physical”.
The definition of Figure 1 is not good and need to be improved.
Line 103: “TFC”, give the full name when the abbreviation first appeared
Line 103: Figure 3 should be “Figure 2” and the subsequent figures should also be adjusted in the serial number.
Lines 103-112: the paragraph should be rewritten according to the results presented as the Figure 3 (should be corrected to Figure 2). For instance, the values of TPC and TFC of major spices, mixed red pepper, and sauté samples should be checked and corrected due to these values were not consistent with the figure. There is no significantly differences in TFC among cardamom, garlic, ginger, black cumin, and the p value should be >0.05.
Axis scale of Y axis of Figure 3 in page 4 should be added.
Line 116: “DPPH, FRAP and FICA”, give the full names when the abbreviations first appeared
Line 118: the punctuation “;”after the word “potentials” should be changed to “,”.
Lines 118-119: Statistical differences in DPPH between cardamom, garlic, ginger and black cumin should be re-defined, because there was no significant differences in DPPH between garlic, ginger and black cumin as shown in Figure 4 in Page 6.
Axis scale of Y axis of Figure 4 in page 6 should be added.
Lines 140: Figure 2 should be changed to Figure 5.
Lines 151-154: “About 20% and 100% of the red pepper samples contained AFB1 levels above the maximum permissible limit of the USA and European Commission, respectively, for spices. However, Ethiopia has not established the residue limit of AFB1 in spices.” The sentence should be moved to the “Discussion” section.
Figure 5 in Page 7: add axis scale in Y axis; the legend of “RP, EP, ES, CP, CS” under the picture is repeated with that under the column pictures, and should be deleted.
Lines 191-198: the paragraph repeats the previous one, delete the paragraph.
Table 2 in Page 7 should be changed to Table 1, and the subsequent Tables should also be adjusted in the serial number.
The conclusion section is a bit long and should be more refined
Lines 425-428: “y = 1.51257x+1.1351, R=0.987 (1) Where; y is the absorbance 427 x is the concentration of AFB1 (ng ml-1)” should be deleted, due to these information are contained in the Figure below.
Author Response
Dear Sir/ Madam;
I hope you are doing well.
As your comments and concerns, all valuable comments are incorporated, you can see from the attached document. Here is the highlight for your convenience.
Response to Reviewer 2 Comments
- Quality of English Language: English spelling and other punctuations, grammar and sentence construction, the style and paragraph arrangement revisited throughout the whole document.
- Introduction/ background and relevant references sufficiency: The background information about spices and spices cooking, and aflatoxin and aflatoxin detoxification is well elaborated and sufficient. The cited publication are the most recent and relevant to the title. From the publications, 2008 cited once, 2011 cited twice, 2012 cited four times, 2015 cited seven times; however, the remaining (77.78%) are recent and published after 2015.
- Adequate description of methods: The methodology of this experiment is revised and well described as of the aflatoxin B1 kit manufacturer’s instruction, particularly section 5.1 (to some extent), 5.2 (partly), 5.3 as a whole and the sequential order of this sub tile is reorganized as 5.3.1 changed to 5.3.2, and vice versa.
- The results clear presentation: The study finding results are re-organized; section 2.1 changed to 2.2 and vice versa. The whole results are more clearly and briefly revised.
- The conclusions: The conclusions are supported by the revised results and expressed in brief.
Response 1 (Point 1): Lines 34-42: The publications cited here are changed, particularly considering the characteristics of aflatoxins in the spices. The publications suggested by the honored reviewer is in vivo and other plant (mushroom) based; however, I tried to read for further knowledge.
Response 2 (Point 2): Line 59: Corrected as commented, ‘easy to be obtained’.
Response 3 (Point 3): Line 63: Corrected as commented, ‘,’ added.
Response 4 (Point 4): Figure 1 modified and defined well; ‘Chemical structure of aflatoxin B1: (a) The numbers indicate carbon atoms in AFB1, C17H12O6; and (b) The number (1-15)’indicate reactive sites of AFB1.
Response 5 (Point 5): Line 103: Corrected as commented, TFC full name is given as it appeared first.
Response 6 (Point 6): Line 103: All figures are rearranged and the results interpreted accordingly. Moreover, the discussion is also done as of the sequential order of the results.
Response 7 (Point 7): Lines 103-112: Some paragraphs including commented one are corrected accordingly based on figure’s serial number, and all significance difference is compared with p0.05
Response 8 (Point 8): Axis scale of Y-axis of figure 3 is incorporated.
Response 9 (Point 9): Line 103: Corrected as commented, DPPH, FRAP and FICA full names are given as they appeared first.
Response 10 (Point 10): Line 118: Corrected as commented, ‘,’ added.
Response 11 (Point 11): Lines 118-119: Antioxidant assays are completely modified based on revised concept including the figure.
Response 12 (Point 12): Axis scale of Y-axis of Figure 4 is incorporated.
Response 13 (Point 13): Line 140: As commented, Figure 2 is changed to Figure 5.
Response 14 (Point 14): Lines 151-154: As commented, the sentence is moved to the ‘discussion’ section and re-written in brief.
Response 15 (Point 15): Axis scale of Y-axis of Figure 5 is corrected and changed to figure 2. Full names of ‘RP, EP, ES, CP, CS’ are given as they appeared first and not repeated under the pictures.
Response 16 (Point 16): Lines 191-198: The paragraph is deleted.
Response 17 (Point 17): Corrected as commented, Table 1, and the subsequent Tables are adjusted in the serial number.
Response 18 (Point 18): The conclusion section is revised and made short.
Response 19 (Point 19): Lines 425-428: Corrected as commented.

Reviewer 3 Report
The paper titled “Aflatoxin B1 Detoxification Potentials of Garlic, Ginger, Cardamom, Black cumin, and Sautéing in Ground Spice Mix of Red Pepper Products” explores the potential of cooking, phytochemicals content and antioxidant activities derived from different spices to detoxify AFB1. The topic is scientifically relevant.
The manuscript is well-structured and clearly written. Tables are clear and well-presented, although some technical corrections need to be made. However, the figures illustrating results need to be explained better (e.g., letters above bars are not explained).
Cited references are relevant and recent.
I would point out that the authors used many abbreviations in the text, therefore, list of abbreviations would contribute to its easier understanding.
Title is suitable and illustrates the aim of the research.
Introduction is concise and informative, indicating the background and the aim if the research.
Results are presented in clear and understandable manner, and comprehensively discussed.
Study design is described in detail, enabling the reader to replicate the research. The method for AFB1 determination is crucial for the obtained results on detoxification, however, the authors have not provided the evidence about the method performance. I suggest to the authors to employ an extensive method validation and provide the reader with the appropriate method performance results (at least recovery and precision data, LOD, LOQ).
Conclusions are derived from the results although could be presented better. The conclusions is written in general matter and need to be more concise and straightforward.
Author Response
Dear Sir/ Madam;
I hope you are doing well.
As your comments and concerns, all valuable comments are incorporated, you can see from the attached document. Here is the highlight for your convenience.
Response to Reviewer 3 Comments
- Quality of English Language: English spelling and other punctuations, grammar and sentence construction, the style and paragraph arrangement revisited throughout the whole document.
- The results clear presentation: The study finding results are re-organized; section 2.1 changed to 2.2 and vice versa. The whole results are more clearly and briefly revised.
- The conclusions: The conclusions are supported by the revised results and expressed in brief.
- Letters above bars: Corrected as commented, all figures illustrating results are explained using letters above bars according to their significant difference at p<0.05.
- Abbreviations: Corrected as commented, abbreviations used more than three time in the document are defined.
- The method performance: Linearity and LOD are evaluated.

Round 2
Reviewer 2 Report
The quality of the revised manuscript has greatly improved, such as English presentation, results analysis and discussion were well revised. However, there are still some issues in the manuscript that need to be revised before accepted.
1. In the section of Introduction (lines 55-59), detrimental properties of AFs should be introduced such as carcinogenic, teratogenic, mutagenic, nephrotoxic and hepatotoxic properties, and allow me to suggest the following publications to be cited herein, please.
Wang, Y.; Liu, F.; Zhou, X.; Liu, M.; Zang, H.; Liu, X.; Shan, A.; Feng, X. Alleviation of Oral Exposure to Aflatoxin B1-Induced Renal Dysfunction, Oxidative Stress, and Cell Apoptosis in Mice Kidney by Curcumin. Antioxidants 2022, 11, 1082. https://doi.org/10.3390/antiox11061082
Loncar, J.; Bellich, B.; Cescutti, P.; Motola, A.; Beccaccioli, M.; Zjalic, S.; Reverberi, M. The Effect of Mushroom Culture Filtrates on the Inhibition of Mycotoxins Produced by Aspergillus flavus and Aspergillus carbonarius. Toxins 2023, 15, 177. https://doi.org/10.3390/toxins15030177
2. Lines 153-154: “The TPC of major spices ranged from 5.77 to 13.06 mg GAE g-1.” and lines 156-157 “The TFC of black cumin, ginger, garlic and cardamom, respectively was 7.92, 2.73, 2.65 and 1.43 mg QE g-1 were significantly (p<0.05) different”. But as shown in the left panel of Fig. 3, all major spices contained TPC or TFC with more than 12 mg GAE g-1 or 12 mg QE g-1. Correct them, please.
3. Both the horizontal and vertical coordinates in figure 3 and 4 are missing. Add them please.
4. The formula in blocks of figure 5 “? =1.51257?+1.1351, ?=0.987” should be deleted, due to it was repeated with the formular in line 408 and seems cumbersome.
Author Response
Response to Reviewer 2 Comments in Round 2
- Quality of English Language: English spelling checks and others are considered throughout the whole document.
- Introduction: The background information about spices and spices cooking, and aflatoxin B1 detoxification through mixing and cooking is sufficiently elaborated. However, the points raised by the honored reviewer about carcinogenic, mutagenic, … property of AFB1 is incorporated in brief with relevant and additional citation (reference [6]).
- The results clear presentation: The study finding results in abstract 65.59 corrected as 65.95 furthermore, section 2.2, and figure 3 and 4 are also corrected as commented.
Response 1 (Point 1): Lines 55-59: In this part, one new paragraph with new citation is incorporated as commented; newly added paragraph: ‘It is capable …. [6].’ The cited reference: Benkerroum, N. Chronic and acute toxicities of aflatoxins: Mechanisms of action. Int. J. Environ. Res. Publ. Health 2020, 17(2), 1-18, which is more relevant for the chapter content. The cited reference here is cross referred from the journal suggested by the honored reviewer; ‘Wang, Y.; Liu, F.; Zhou, X.; Liu, M.; Zang, H.; Liu, X.; Shan, A.; Feng, X. Alleviation of Oral Exposure to Aflatoxin B1-Induced Renal Dysfunction, Oxidative Stress, and Cell Apoptosis in Mice Kidney by Curcumin. Antioxidants 2022, 11, 1082. https://doi.org/10.3390/antiox11061082’
Response 2 (Point 2): Line 153-154 and line 156-157 are corrected as commented.
Response 3 (Point 3): Figure 3 and 4 graphics is changed, both horizontal and vertical coordinates are incorporated.
Response 4 (Point 4): The formula expressed as equation 1 deleted to avoid repetition, so numerically expressed eq. 1, eq. 2 are also corrected accordingly.
